# Development of Hydrophobic-Modified Nanosilica for Pressure Reduction and Injection Increase in an Ultra-Low-Permeability Reservoir

Hao Lai [1,2,3], Wei Shi [1,2,*], Junqi Wang [3], Lei Tang [4] and Nanjun Lai [1,2,3,*]

1   School of Chemistry and Chemical Engineering, Southwest Petroleum University, Chengdu 610500, China
2   Oil & Gas Field Applied Chemistry Key Laboratory of Sichuan Province, Chengdu 610500, China
3   The Key Laboratory of Well Stability and Fluid & Rock Mechanics in Oil and Gas Reservoir of Shaanxi Province, Xi'an Shiyou University, Xi'an 710065, China
4   Sichuan Ruidong Technology Co., Ltd., Chengdu 610500, China
*   Correspondence: shiwei80@swpu.edu.cn (W.S.); lainanjun@126.com (N.L.)

**Abstract:** A low-permeability reservoir contains many fine pore throat structures, which result in excessive injection pressure of the water injection well and difficult water injection in the production process of a low-permeability reservoir. In this study, a new silane coupling agent was synthesized via the ring-opening reaction between dodecyl amine and KH-560 ($\gamma$-propyl trimethoxysilane). The modified KH-560 was reacted with nano-$SiO_2$ to synthesize the modified nano-$SiO_2$ as an antihypertensive additive. Fourier infrared spectroscopy, thermogravimetric analysis and laser scattering were used to characterize this modified nano-$SiO_2$. The results show that the particle size of the modified nano-$SiO_2$ is less than 60 nm. The test results of the water contact angle show that the dispersion system can increase the rock contact angle from 37.34° to 136.36°, which makes the rock surface transform from hydrophilicity to hydrophobicity and reduce the binding effect of rock with water. The dispersion test shows that the modified nano-$SiO_2$ has good dispersion stability under alkaline conditions with TX-100 (Polyethylene glycol octylphenyl ether) as the dispersant. The antiswelling test results show that the antiswelling rate of this modified nano-$SiO_2$ is 42.9%, which can efficiently prevent the clay expansion in the formation to reduce the injection pressure. The core displacement test results show that its depressurization rate reaches 49%. The depressurization rate still maintains 46% at a 20 PV water flow rate, indicating that its depressurization effect is remarkable and it has excellent erosion resistance.

**Keywords:** nano-$SiO_2$; hydrophobicity; ultra-low-permeability reservoir; pressure reduction

## 1. Introduction

In recent years, given the newly added geological reserves in China, low-permeability and very-low-permeability geological reserves have increased yearly [1,2]. Based on previous statistical results, the porosity of low-permeability reservoirs is less than 15% and their permeability is less than 20 mD [3]. Many fine pore throat structures exist in low-permeability reservoirs [4]. Owing to the existence of microcapillary forces, the solid–liquid interface and the electrochemistry effect, a start-up pressure difference exists in low-permeability reservoirs [5,6]. Moreover, the clay in the formation is easy to expand [7,8], which leads to the high pressure of the water injection well. So, it directly affects the efficiency of water injection, increases the burden of the water injection system and causes high energy consumption. In addition, a casing rupture will occur when the water injection system is under a long-term high-pressure condition, which would result in a reduction in sweep efficiency and oil recovery efficiency [9]. Therefore, the development of high-efficiency pressure-reducing and injection-enhancing agents is key to the exploitation of low-permeability reservoirs.

The pressure-decreasing and augmented injection in low-permeability reservoirs have been extensively studied. At present, the technology of pressure-decreasing and augmented injection has two main methods: reservoir modification and surface modification [10]. Reservoir reconstruction technology is used to expand the effective diameter of rock pores by acidizing plugging removal and hydraulic fracturing [11,12] to reduce injection resistance to achieve the result of reduced pressure and increased injection [13–15]. In the process of hydraulic fracturing and acidification, it is easy for the reservoir to be injured by the intrusion of external fluids, resulting in a serious decline in the exploitation degree [16–18]. Therefore, how to reduce the damage to the reservoir and the equipment during the use of reservoir reconstruction technology is also a main problem that needs to be solved urgently. Surface modification technology is utilized to change the rock surface properties of pores [19] and prevent clay from swelling by injecting modifiers into rock pores [20], so as to achieve the effect of reducing pressure and increasing injection [21–24].

Hydrophobically modified nano-$SiO_2$, as a modifier to reduce pressure and increase injection, is a newly developing technology. The hydrophobic nano-$SiO_2$ dispersion is injected into the formation to achieve the pressure-decreasing and augmented injection of the reservoir. The surface of the modified nano-$SiO_2$ particles has unsaturated bonds and strong adsorption capacity. After entering the formation, the nano-$SiO_2$ adsorbs onto the surface of the rock pores, replacing the existing hydration film and forming a new adsorption layer, making the rock pores hydrophobic. It reduces the flow resistance and the contact between the water and the rock pores, achieving the effect of the pressure-decreasing and augmented injection [25,26]. Liu [27] synthesized an environmentally responsive modified nano-$SiO_2$, which was covalently modified by organic compounds containing hydrophobic groups and double bonds. The chemical adsorption method was used to cover a layer of hydrophilic organic matter on the surface of the modified nano-$SiO_2$ to ensure its good dispersibility in water. Owing to the change in the environment (temperature, pH and salinity), the nano-$SiO_2$ would be released from the dispersion and adsorbed onto the rock pores when the nano-$SiO_2$ enters the injection well from the normal environment, thereby achieving the effect of reduced pressure and increased injection. Dai [28] used dimethyldichlorosilane to modify nano-$SiO_2$. The core-flow experiment shows that the modified nano-$SiO_2$ exerts a better effect on depressurization and injection in low-permeability reservoirs, and the depressurization rate reaches 45%. Zhao [29] used n-propyl trichlorosilane to modify nano-$SiO_2$ and evaluated the effect of the pressure-decreasing and augmented injection through core-flow experiments. The results showed that the depressurization rate of the modified nano-$SiO_2$ in ultra-low permeability reservoirs reached 31.47%. A roughness reduction mechanism was proposed to explain their pressure reduction mechanism. Hydrophobic silica nanoparticles can adsorb onto the sand surface, reduce the roughness of the flow channel, partially change the wettability of the flow channel and reduce its water wettability. Flow resistance and injection pressure are reduced accordingly.

The above-mentioned pressure-reducing and injecting agent has an unsatisfactory depressurization effect under the condition of ultra-low-permeability (less than 10 mD). Based on this problem, a new modified nano-$SiO_2$, used as a depressurization and injection enhancement agent, was synthesized in this study, which has excellent pressure-decreasing and augmented injection effects under the formation condition with permeability less than 10 mD. The structure and particle size of prepared nano-$SiO_2$ were characterized using infrared spectrometry and dynamic light scattering. The wettability, thermal stability, dispersion stability and antiswelling properties of the modified nano-$SiO_2$ were simultaneously studied in this study. The performance of the pressure-decreasing and augmented injection of modified nano-$SiO_2$ was evaluated via an indoor core displacement experiment. This study hoped to provide a new modification strategy for solving the problem of excessive injection pressure in low-permeability reservoirs.

## 2. Experimental Sections

### 2.1. Experimental Materials and Instruments

Materials: nano-$SiO_2$, Shanghai Aladdin Biochemical Technology Co., Ltd.; lauryl amine, AR, Shanghai Xianding Biological Technology Co., Ltd.; $\gamma$-propyl trimethoxysilane (KH560), BR, Shanghai Yuanye Bio-Technology Co., Ltd.; sodium hydroxide (NaOH), absolute ethanol ($C_2H_5OH$) are both AR, Chengdu Kelong Chemicals Co., Ltd.; Polyethylene glycol octylphenyl ether (TX-100), CP, Chengdu Kelong Chemicals Co., Ltd.; kerosene, Unified Petrochemical Co., Ltd.; bentonite, Chengdu Hua Chemical Reagent Co., Ltd.; artificial core.

Instruments: heat-collecting-type constant-temperature-heating magnetic stirrer, Beijing Kewei Yongxing Instrument Co., Ltd.; Analytical Balances, Mettler-Toledo Instruments Co., Ltd.; constant temperature drying oven, Chengdu Test Instrument Co., Ltd.; ultrasonic cleaner, Shanghai Kedao Ultrasonic Instrument Co., Ltd.; centrifuge, Shanghai Medical Devices Co., Ltd.; rotary evaporator, Gongyi Yuhua Instrument Co., Ltd.; infrared spectrometer, Thermo Electron Corporation; contact angle measuring instrument, German Kruss Scientific Instruments Shanghai Co., Ltd.; synchronous integrated thermal analyzer, Netzsch Group; zeta potential analyzer, American CD Company; laser scattering system, Brookhaven Instruments, Inc.

### 2.2. Experimental Method

#### 2.2.1. Preparation of Modified Nano-$SiO_2$

Owing to the hydrophobicity of a long carbon chain, it is often used to endow hydrophobicity to nano-materials. Wang [30] declaimed that the ring-opening reaction could occur between amino groups and epoxy groups. In this article, the silane coupling agent (KH560) was used as an intermediate to connect a long carbon chain and nano-$SiO_2$.

Specifically, dodecyl amine was utilized as a modifier to graft long carbon onto the KH560, and then the modified KH560 containing a long carbon chain was branched to nano-$SiO_2$. After the experiment mentioned above, the nano-$SiO_2$ with hydrophobicity was prepared. The experimental steps were as follows.

(1) Dodecyl amine (2.2242 g) was dissolved in 10 mL of absolute ethanol, and then it was placed into a three-hole round-bottom flask and stirred in a thermostatic heating magnetic stirrer at 60 °C. At the same time, 2.83 g of KH560 was added into the three-hole round-bottom flask drop by drop and it reacted for 2 h. Then, the cooled dispersion liquid was placed in a rotary evaporator at 50 °C for 20 min until the ethanol rotary evaporation was complete. Finally, the viscous oil-like liquid (3.67 g) was obtained, which was modified KH-560. Additionally, the yield was 72.6%.

(2) About 0.8 g of nano-$SiO_2$ was dried at 120 °C for 2 h. Then, it was added to 20 mL of absolute ethanol. The above liquid was ultrasonically processed in an ultrasonic cleaner for 30 min to ensure the uniform dispersion of the nano-$SiO_2$. Subsequently, 1.6 g of modified KH560 was dissolved in 20 mL of absolute ethanol and stirred evenly, and it was added to the nano-$SiO_2$ dispersion slowly. The mixture was stirred in a constant-temperature-heating magnetic stirrer for 10 min to mix it evenly. Then, the thermostatic heating magnetic stirrer was adjusted to 60 °C, and the thermostatic reaction continued for 4 h under the condition of nitrogen gas. The reaction dispersion liquid was centrifuged at a high speed for 5 min and washed with absolute ethanol three times until the remaining modified KH560 was removed. The precipitation in the centrifugal tube was taken and dried for 4 h in a drying oven at 120 °C. Finally, the modified nano-$SiO_2$ was obtained.

#### 2.2.2. Infrared-Spectroscopy Analysis

A Fourier infrared spectrometer was used to characterize the modified nano-$SiO_2$. Small amounts of modified nano-$SiO_2$ and KBr were mixed uniformly and pressed into tablets after grinding. Then, they were placed in an infrared spectrometer to obtain the

infrared spectrum of modified nano-$SiO_2$. The measured wavenumbers ranged from $0$ cm$^{-1}$ to $4000$ cm$^{-1}$, the wavenumber precision was $0.01$ cm$^{-1}$ and the resolution was $4$ cm$^{-1}$.

### 2.2.3. Wettability of Modified Nano-$SiO_2$

The wettability of modified nano-$SiO_2$ was evaluated by measuring the contact angle with the German KRUSS DSA30S interface parameter-measuring instrument (Shanghai KRUSS Scientific Instrument Co., Ltd., Shanghai, China). First, the modified nano-$SiO_2$ with different concentration gradients was dispersed in TX-100. Subsequently, the core was cut into thin slices (3 mm) and soaked in the prepared mixture for 24 h. Then, the prepared sample was put into a constant-temperature drying oven and dried at 120 °C. Finally, the contact angle of the water droplet and core was measured using an interface surface parameter-measuring instrument.

### 2.2.4. Dynamic Light Scattering Test

The particle size of the modified nano-$SiO_2$ was analyzed via dynamic light scattering. The particle size distribution was measured using a laser scattering system from Brookhaven Instruments, USA. First, the modified nano-$SiO_2$ was dispersed in TX-100, and then the particle size distribution of modified nano-SiO2 was obtained via dynamic light scattering.

### 2.2.5. Thermogravimetric Analysis (TGA)

The TGA of the nano-$SiO_2$ and modified nano-$SiO_2$ was performed using a thermogravimetric analyzer from 0 °C to 1000 °C. Under nitrogen flow, the samples were continuously heated to 1000 °C with a step size of 10 °C/min, and the data of the change in sample weight with temperature were collected. By analyzing the weight change in the samples at each stage, the amount of groups grafted onto the nano-$SiO_2$ was quantitatively evaluated.

### 2.2.6. Zeta Potential Test

Equal amounts of modified nano-$SiO_2$ were dispersed in TX-100 and NaOH-regulated TX-100, respectively. The zeta potential of the two dispersions was measured using a Zeta PALS 190 Plus Potential Analyzer (Brookhaven Instruments, Austin, TX, USA). Zeta potential was used to evaluate the stability of the two dispersions.

### 2.2.7. Dispersibility and Stability

The modified nanosilica was dispersed in water, TX-100, ethanol, kerosene and NaOH-adjusted TX-100. The dispersion conditions of the five kinds of dispersion liquids were observed. After storing them for 30 days, the dispersion conditions of the five kinds of dispersion liquids were observed again to judge the stability of the dispersion liquids.

### 2.2.8. Antiswelling Test

The antiswelling rate of modified mano-$SiO_2$ was evaluated using the centrifugal method. The antiswelling ratio was evaluated by measuring the swelling volume of bentonite in kerosene, modified nano-$SiO_2$ dispersions and pure water. Bentonite (1 g) was added into 20 mL of kerosene, water and modified nano-$SiO_2$ dispersions, respectively, before shaking at room temperature for 2 h. After storing it for 24 h, the swelling volume of bentonite was recorded.

The antiswelling rate was calculated as follows:

$$B_1 = \frac{V_2 - V_1}{V_2 - V_0} \times 100\% \tag{1}$$

where $B_1$ is the antiswelling rate, %; $V_0$ is the swelling volume of bentonite in kerosene, mL; $V_1$ is the swelling volume of bentonite in modified nano-$SiO_2$ dispersion, mL, and $V_2$ is the swelling volume of bentonite in pure water volume, mL.

### 2.2.9. Evaluation of the Effect of Blood Pressure Reduction and Injection

The prepared physical parameters of the low-permeability cores are shown in Table 1.

**Table 1.** Physical parameters of core.

| Length (cm) | Diameter (cm) | Porosity (%) | Permeability ($10^{-3}$ $\mu m^2$) | Dry Weight (g) |
|---|---|---|---|---|
| 10 | 2.5 | 9.4 | 2.409 | 78.47 |

The confining pressure of the core was set as 8 MPa, and the core was vacuum dried at 100 °C for 24 h. First, the mass ($W_1$) of the dry core was measured, and it was immersed in water for 24 h. Then, the mass ($W_2$) of the core under a wet state was measured. As shown in Figure 1, the pipeline of the displacement experiment was connected. Brine was injected into the core at a constant flow rate of 0.1 mL/min, and the displacement pressure was recorded until the pressure $P_1$ stabilized. Water permeability was then calculated. The modified nano-SiO$_2$ dispersion of 0.5 PV was injected at a constant flow rate of 0.1 mL/min and aged at 60 °C for 48 h. During the subsequent water flooding, brine was injected at a constant flow rate of 0.1 mL/min until the pressure was stable after water flooding, and the stable pressure $P_2$ was recorded. Subsequently, 30 PV of brine was injected at a constant flow rate of 0.1 mL/min to flush the core, and the pressure during displacement was recorded to evaluate the depressurization and boosting effect as well as the erosion resistance of the modified nano-SiO$_2$.

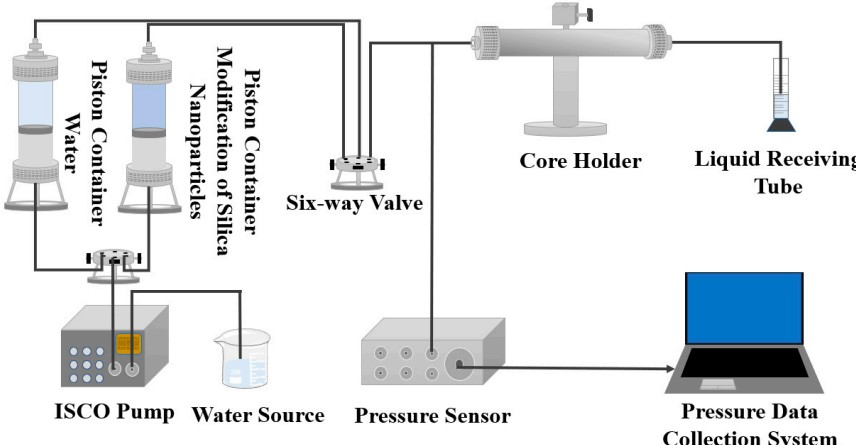

**Figure 1.** Physical model of displacement experiment.

The core parameters were calculated as follows:

$$PV = \frac{W_1 - W_2}{\rho} \tag{2}$$

$$k = \frac{Q\mu l}{\Delta p A} \tag{3}$$

where $\rho$ is the density of water, g/cm$^3$; $Q$ is the flow rate of fluid passing through the core per unit time, cm$^3$/s; $\mu$ is the viscosity of liquid, mPa/s; $l$ is the core length, cm; $\Delta p$ is the pressure difference before and after the liquid passes through the core, MPa, and $A$ is the cross-sectional area of the liquid passing through the core, cm$^2$.

## 3. Results and Analysis

### 3.1. Characterization of Modified Nano-SiO$_2$

The FR-IR spectra of nano-SiO$_2$, modified KH-560 and modified nano-SiO$_2$ are presented in Figure 2. The characteristic peaks at 472 and 1105 cm$^{-1}$ belong to the bending

vibration and stretching vibration of the Si-O. The antisymmetric tensile vibration absorption peak and bending vibration absorption peak of the hydroxyl emerged at 3446 and 1633 cm$^{-1}$. Compared with the data of nano-SiO$_2$, the peak intensity at 3446 cm$^{-1}$ of the modified nano-SiO$_2$ decreased, indicating that the hydroxyl on the nano-SiO$_2$ had reacted. In addition, the characteristic peaks of methyl and methylene can be found at 2925 and 2858 cm$^{-1}$. The two characteristic peaks mentioned above can be simultaneously observed in the spectra of modified nano-SiO$_2$ and modified KH-560, indicating that the modified KH-560 successfully grafted onto the modified nano-SiO$_2$.

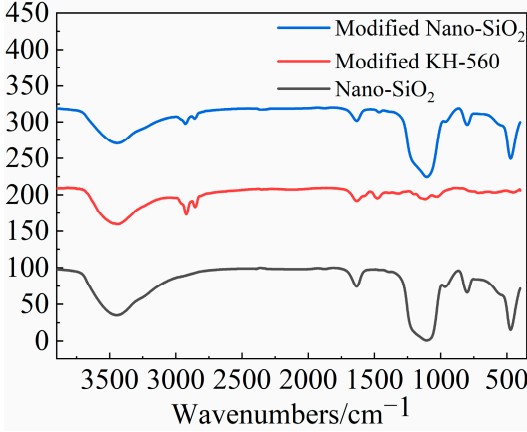

**Figure 2.** Infrared spectra of nano-SiO$_2$ before and after modification.

### 3.1.1. Dynamic Light Scattering

Nano-SiO$_2$ has strong hydrophilic properties, and it easily agglomerates between particles, thereby enlarging the particle radius. Figure 3 displays the particle size distribution of modified nano-SiO$_2$. All of the particle sizes of the modified nano-SiO$_2$ were less than 60 nm, and the median particle size was 15 nm, in line with the nanoparticle size range. The modified nano-SiO$_2$ had less agglomeration, and the obtained particles had smaller and uniform particle size. When it was injected into low-permeability oil reservoirs, it was not easy to block and it more easily adsorbed onto the surface of rock pores evenly.

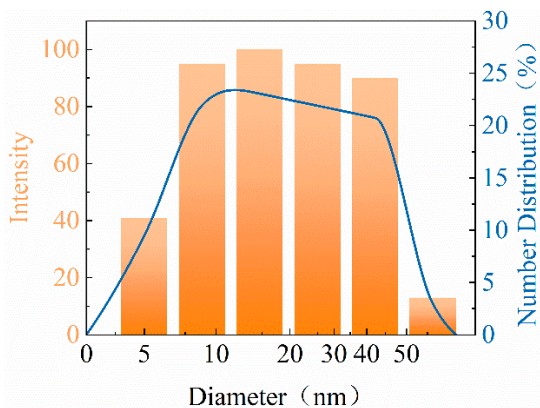

**Figure 3.** Particle size distribution.

### 3.1.2. TGA

Figure 4 shows the TGA of SiO$_2$ before and after modification. With increased temperature, the weight of the sample gradually decreased, and the mass change with temperature was divided mostly into two stages. In the first stage, with increased temperature from 25 °C to 120 °C, the mass of the sample decreased from 100% to 98%, which was primarily caused by the loss in adsorbed water on the sample surface. In the second stage, with increased temperature from 120 °C to 1000 °C, the weight of nano-SiO$_2$ decreased from

98% to 92%, which was due to the weight change caused by the dehydration and polycondensation of the hydroxyl groups on the nano-SiO$_2$ surface. However, the weight of the modified nano-SiO$_2$ decreased from 98% to 89% with increased temperature. On the one hand, this was due to the dehydration and polycondensation of the unreacted hydroxyl groups on the surface of the modified nano-SiO$_2$, the same as with the above-mentioned SiO$_2$. On the other hand, the weight of the modified KH560 grafted onto the surface of SiO$_2$ was further reduced due to the fracture of the modified KH560 owing to the excessive temperature. At 1000 °C, the weight of SiO$_2$ before and after modification differed by 3%, indicating that 3% of KH560 was grafted onto the surface of SiO$_2$.

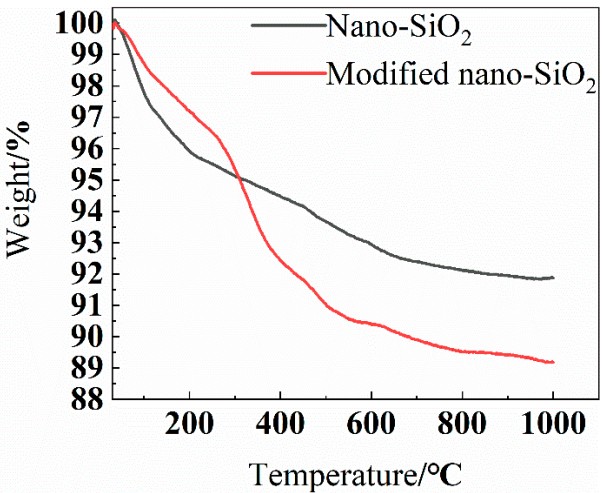

**Figure 4.** Thermogravimetric curve.

*3.2. Performance Evaluation of Modified Nano-SiO$_2$*

3.2.1. Contact Angle

Wettability is very important as it directly affects the effect of pressure reduction and injection. Core wettability was evaluated by measuring the contact angle. Figure 5 shows the contact angles of core slices immersed in modified nano-SiO$_2$ dispersion and saline. In the control experiment of core sections soaked in salt water, the core contact angle was 37.34°, indicating that the initial surface wettability of the core was hydrophilic. However, the contact angle of the core slices immersed in the modified nano-SiO$_2$ dispersion solution was 104.26°, indicating that the modified nano-SiO$_2$ can change the core from hydrophilic wetting to hydrophobic wetting, and the change in wettability can reduce the adhesion force of the rock surface to water to promote water flow. Consequently, the effect of lowering the pressure and increasing injection was achieved. With an increased concentration of modified nano-SiO$_2$, the contact angle increased from 104.26° to 136.36° and gradually became stable. This finding indicated that with increased concentration, the adsorption amount on the core surface increased, and the hydrophobic degree on the core surface also increased. However, with a further increased concentration of modified nano-SiO$_2$, the adsorption gradually became saturated. The adsorption capacity was basically unchanged and the contact angle tended to be stable.

3.2.2. Stability and Zeta Potential

Modified nano-SiO$_2$ was dispersed in pure water, TX-100, ethanol and kerosene to explore the dispersibility of nano-SiO$_2$. As shown in Figure 6, due to the large specific surface area and high surface energy of modified nano-SiO$_2$, it was unable to disperse in water and agglomerate seriously. In ethanol solution, the dispersion effect was poor, and some modified nano-SiO$_2$ agglomerated and precipitated. In TX-100, the dispersion effect of modified nano-SiO$_2$ was better, but the solution was cloudy. The nano-SiO$_2$ had good dispersibility in the kerosene dispersion system, and the solution was clear and transparent.

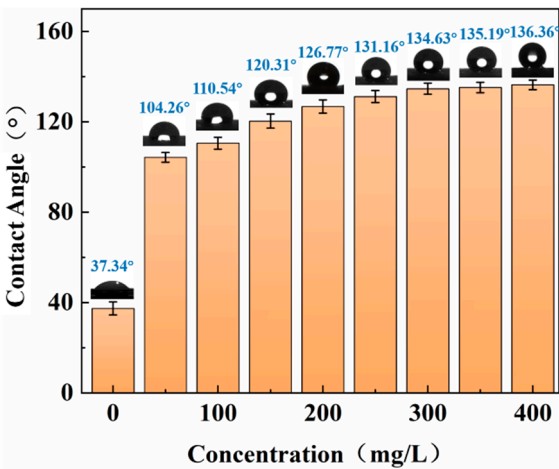

**Figure 5.** Contact angle of core before and after adsorption of modified nano-SiO$_2$.

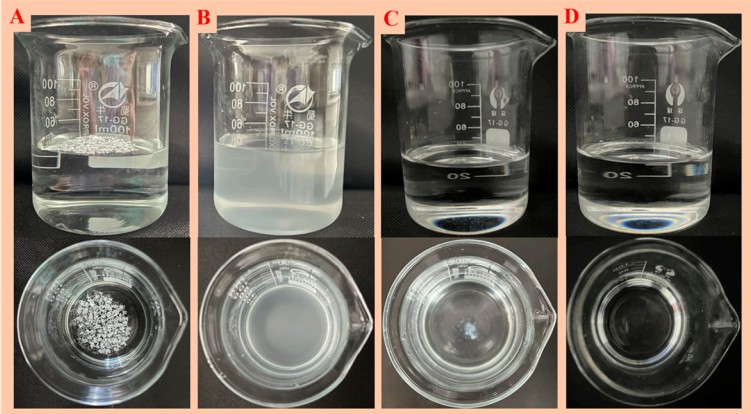

**Figure 6.** Modified nano-SiO$_2$ dispersion in water (**A**), Polyethylene glycol octylphenyl ether (TX-100) (**B**), ethanol (**C**) and kerosene (**D**).

To explore the dispersion stability, the dispersion liquid was stored for 30 days, and then the solution dispersion was observed. Figure 7 shows the dispersion situation of the above-mentioned solution. Compared with the dispersion solution not stored for 30 days, precipitation occurred in the surfactant aqueous dispersion system. The precipitation in the ethanol dispersion system increased significantly, but the kerosene dispersion system remained well dispersed.

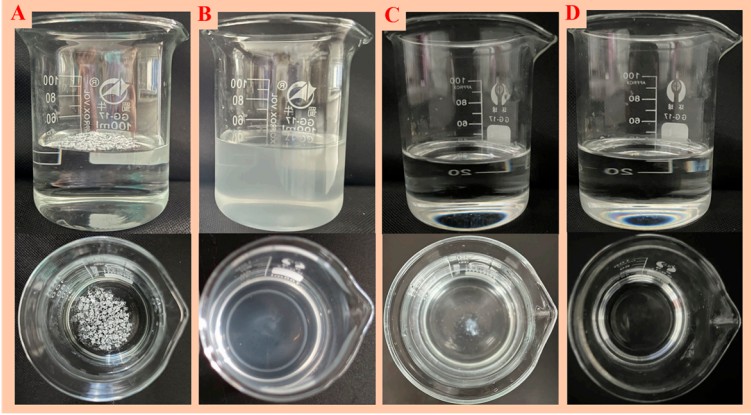

**Figure 7.** State of modified nano-SiO$_2$ after 30 days of dispersion in water (**A**), Polyethylene glycol octylphenyl ether (TX-100) (**B**), ethanol (**C**) and kerosene (**D**).

A large amount of working fluid is generally used on site; so, using kerosene as a dispersant is costly and has high safety risks during storage and transportation. Therefore, the dispersion system of adding NaOH to the surfactant aqueous solution needed to be explored. As shown in Figure 8, the dispersion of the aqueous surfactant dispersion system without NaOH was turbid, whereas the dispersion of the aqueous surfactant dispersion system with NaOH was well dispersed, and the dispersion was clear and transparent. Then, the dispersion before and after NaOH adjustment was stored for 30 days. As shown in Figures 7 and 8, the precipitation formed in the dispersion system without NaOH, whereas the dispersion with NaOH remained clear and transparent.

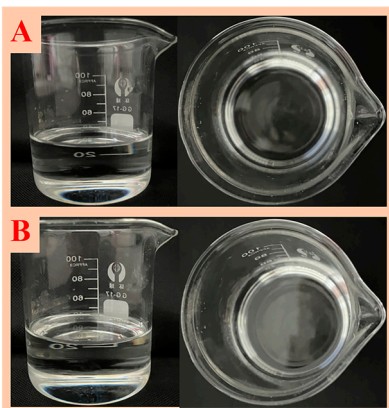

**Figure 8.** Dispersion state of modified nano-$SiO_2$ in NaOH-regulated Polyethylene glycol octylphenyl ether (TX-100) (**A**) and after 30 days of dispersion (**B**).

Nano-$SiO_2$ dispersion is a colloidal system, and zeta potential is also an important parameter affecting its stability. In addition to the stability experiments mentioned above, the zeta potentials of the surfactant aqueous dispersions before and after the introduction of NaOH were also measured. The larger absolute value of the zeta potential indicated that the system was more stable. As shown in Figures 9 and 10, after NaOH addition, the zeta potential of the system increased from $-24.1$ mV to 62.7 mV. Analysis revealed that surfactant TX-100 can form a steric hindrance effect through adsorption onto the surface of modified nano-$SiO_2$, which can prevent mutual collision and coalescence between particles and improve the system stability. After adding NaOH, $OH^-$ adsorbed onto the surface of the modified nano-$SiO_2$ through hydrogen bonds, whereas $Na^+$ was arranged around the modified nano-$SiO_2$ via diffusion, forming a diffusion electric double layer. Consequently, the charge on the surface of nano-$SiO_2$ increased, which improved the repulsion between particles and the system stability.

### 3.2.3. Antiswelling Test

The water expansion of bentonite in the rock channel reduced the effective diameter of the channel and increased the injection pressure. The antiswelling performance is also an important index affecting the influence of the modified nano-$SiO_2$ dispersion under pressure reduction and injection. The swelling volumes of bentonite in kerosene, nano-$SiO_2$ dispersion and water are, respectively, shown in Table 2. The swelling volume of bentonite in kerosene was 1 mL, the swelling volume of bentonite in nano-$SiO_2$ was 1.8 mL and the swelling volume of bentonite in water was 2.4 mL. According to Equation (1), the antiswelling rate of modified nano-$SiO_2$ was 42.9%, and the modified nano-$SiO_2$ had a good antiswelling effect. After analysis, given that the modified nano-$SiO_2$ was hydrophobic, the modified nano-$SiO_2$ was adsorbed onto the surface of bentonite, and a hydrophobic layer formed on the bentonite surface, blocking the contact between water and bentonite to achieve the effect of antiswelling.

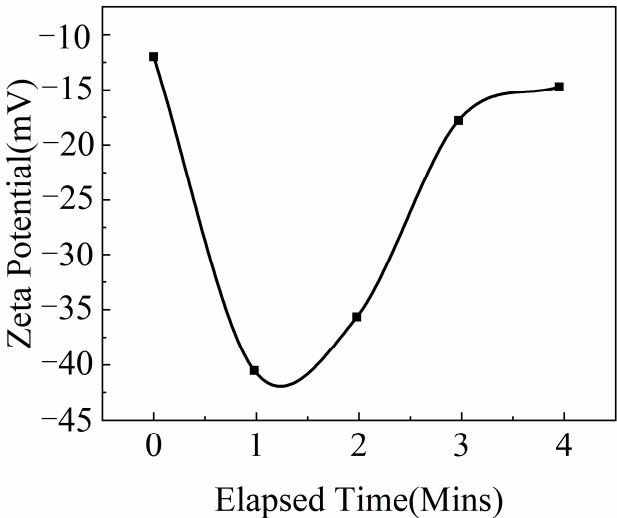

**Figure 9.** Zeta potential of surfactant aqueous dispersion before NaOH conditioning.

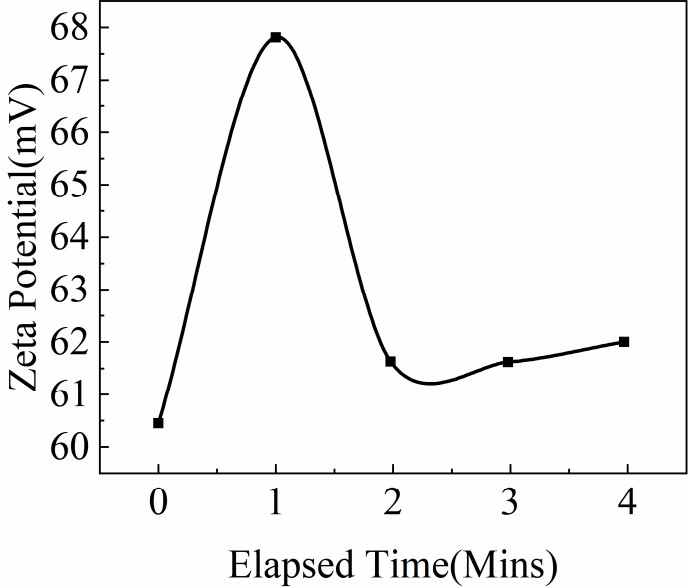

**Figure 10.** Zeta potential of surface active aqueous dispersion after NaOH adjustment.

**Table 2.** Swelling volume of bentonite in different solutions.

| Solution Type | Expansion of the Volume (mL) | The Expansion Rate (%) |
|---|---|---|
| Kerosene | 1 | |
| Modified nano-SiO$_2$ | 1.8 | 42.9 |
| Water | 2.4 | |

### 3.2.4. Depressurization and Injection Enhancement Performance

During the start-up process of water injection, given that the low-permeability core needed to overcome a certain start-up pressure, the pressure initially increased to the maximum value and then decreased to the equilibrium value. The low-permeability core surface had a layer of hydration boundary, and fluid flow was not easy. Smaller core permeability corresponded with a greater resistance to overcome the fluid flow and a greater starting pressure gradient. As shown in Figure 11, the initial injection pressure of the core was 0.141 MPa. After the injection of 0.5 PV nano-SiO$_2$ dispersion solution with a concentration of 300 mg/L, the pressure of the subsequent water flooding was stable at 0.071 MPa, and the pressure reduction rate reached 49.6%. After flushing 20 PV water at

a flow rate of 0.1 mL/min, the displacement pressure was 0.076 MPa, and the pressure reduction rate remained as high as 46%. This result indicated that the modified nano-SiO$_2$ adsorbed onto the surface of the rock passage and can maintain a low injection pressure for a long time even after long-term water erosion, indicating long-term effective performance.

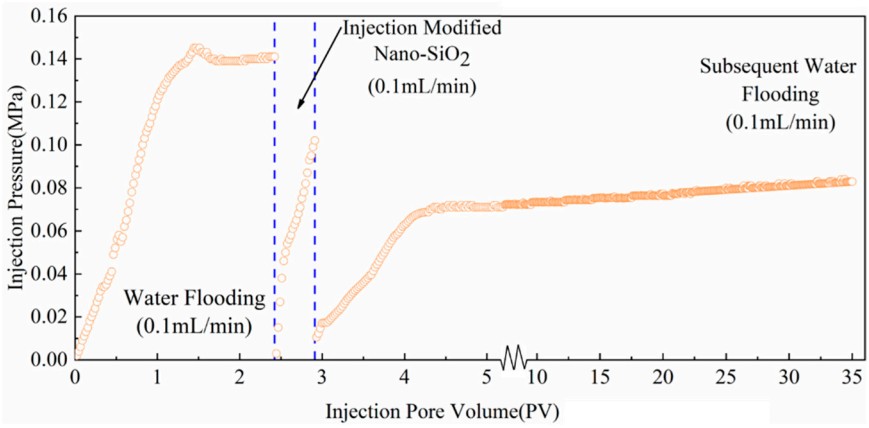

**Figure 11.** Effect of modified nano-SiO$_2$ on water injection pressure.

## 4. Analysis of Mechanism of Reduced Blood Pressure and Increased Injection
### 4.1. Reduced Solid–Liquid Adhesion

The adhesion of solid to liquid reflects the retention ability of the solid surface to liquid. The work of adhesion is the maximum work performed outwards by the system during solid–liquid adhesion. Greater adhesion performance corresponded with greater system stability and a stronger solid–liquid interface. Therefore, reducing the adhesion work of the solid–liquid interface can reduce the resistance of liquid flowing through solid surface. In 1952, Harkins [31] discussed the interface problem from the view of adhesion work. From the perspective of thermodynamics, he introduced the following solid–liquid adhesion work equation:

$$W_a = \gamma_{gl} + \gamma_{gs} - \gamma_{sl} \tag{4}$$

Combined with the Young equation, it is as follows:

$$\gamma_{gs} = \gamma_{gl}\cos\theta + \gamma_{sl} \tag{5}$$

It follows from Equations (4) and (5) that

$$W_a = \gamma_{gl}(1 + \cos\theta) \tag{6}$$

where $W_a$ is the solid–liquid work of adhesion, mJ/m$^2$; $\cos\theta$ is the contact angle of a liquid on a solid surface, °; $\gamma_{gl}$ is the gas–liquid interfacial tension, mN/m; $\gamma_{gs}$ is the gas–solid interfacial tension, mN/m, and $\gamma_{sl}$ is the solid–liquid interfacial tension, mN/m.

According to Equation (6), the adhesion work between a liquid and solid can be obtained only by measuring the surface tension of liquid and the contact angle of liquid on the solid surface, as shown in Table 3. According to the results of wettability evaluation in Section 3.2.1, the solid–liquid adhesion work can be calculated.

**Table 3.** Effect of modified nano-SiO$_2$ on solid–liquid adhesion.

|  | Surface Tension (mN·m$^2$) | Contact Angle (°) | Work of Adhesion (mJ·m$^2$) |
|---|---|---|---|
| before the adsorption | 72.14 | 37.34 | 129.49 |
| after the adsorption | | 134.63 | 21.46 |

The modified nano-$SiO_2$ induced a change in the rock surface from hydrophilic to hydrophobic. The contact angle increased, and the adhesion work of the rock to the water phase decreased from 129.49 mJ/m$^2$ to 21.46 mJ/m$^2$. These phenomena reduced the ability of the rock surface to bind to the water film and increased its fluidity and effective flow space. Thus, the injection capacity of the water phase improved and the injection pressure was reduced.

### 4.2. Prevention of Clay Swelling

The modified nano-$SiO_2$ also exerted an antiswelling effect. As shown in Figure 12, when water flowed through the clay minerals on the surface of the rock pores, the clay particles absorbed water and expanded, thereby reducing the effective diameter of the rock pores and greatly increasing the injection pressure. As shown in Figure 12, after injecting the modified nano-$SiO_2$, the modified nano-$SiO_2$ adsorbed onto the surface of the clay mineral to form a hydrophobic layer and isolate the contact between the water and the clay particles, greatly reducing the probability of hydration expansion of the clay particles and increasing the effective diameter of the rock pores. Accordingly, reduced injection pressure and increased injection volume were also achieved.

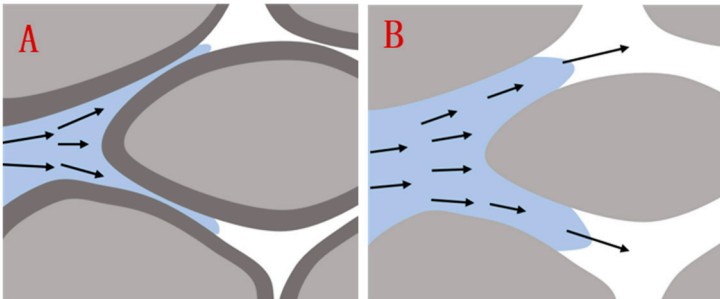

**Figure 12.** Rock pore state before (**A**) and after (**B**) the adsorption of nano-$SiO_2$.

## 5. Conclusions

For solving the problem of excessive high pressure in ultra-low permeability reservoirs, a new pressure-decreasing and augmented injection agent (modified nano-$SiO_2$) was prepared. The structure and particle size of the prepared nano-$SiO_2$ were characterized via infrared spectrometry and dynamic light scattering. Additionally, the test results showed that the particle size of modified nano-$SiO_2$ was less than 60 nm, which contributed to making it pass through smaller cracks. Based on the water contact angle test results, the surface of the rock transformed from having hydrophilicity to hydrophobicity due to the modified nano-$SiO_2$, which led to the weakened binding effect of the rocks with water to prevent the clay expansion of the formation. Under the ultra-low permeability condition, the depressurization efficiency of the modified nano-$SiO_2$ could reach 49%. Additionally, it could be maintained at 46% at 20 PV water flow.

**Author Contributions:** Conceptualization, H.L. and W.S.; methodology, H.L.; validation, H.L. and N.L.; formal analysis, L.T.; investigation, H.L.; resources, N.L.; data curation, J.W.; writing—original draft preparation, H.L.; writing—review and editing, H.L. and W.S.; visualization, H.L.; supervision, J.W.; project administration, N.L.; funding acquisition, N.L. All authors have read and agreed to the published version of the manuscript.

**Funding:** This research received no external funding or This research was funded by [The project was supported by the Opening Fund of The Key Laboratory of Well Stability and Fluid and Rock Mechanics in Oil and Gas Reservoir of Shaanxi Province; The Opening Project of Oil and Gas Field Applied Chemistry Key Laboratory of Sichuan Province] grant number [no. WSFRM20210402001; no. YQKF202010].

**Institutional Review Board Statement:** This study did not require ethical approval.

**Informed Consent Statement:** Not applicable.

**Data Availability Statement:** No new data were created or analyzed in this study. Data sharing is not applicable to this article.

**Conflicts of Interest:** The authors declare no conflict of interest.

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
