# Peer review of "Development of Hydrophobic-Modified Nanosilica for Pressure Reduction and Injection Increase in an Ultra-Low-Permeability Reservoir"

_applsci, doi:10.3390/app13095248_

Round 1
Reviewer 1 Report
I have reviewed the manuscript with the title “Development of Hydrophobic Modified Nanosilica for Pressure Reduction and Injection Increase in An Ultra-Low-Permeability reservoir”. The topic of the manuscript falls within the scope of the journal. This was a well-written paper with some interesting results. The flow of the paper was logical, the experiments were well-described, and the paper was well-referenced. The data presented is convincing. The following are my comments
1. The English language needs some improvement
2. The novelty and contribution of this work shall be outlined properly in the introduction section.
3. An abstract can be improved by adding results quantitatively.
4. Keywords can be improved
5. The objective and clear motivation of the manuscript is not well defined.
6. The introduction can be enhanced by doing more literature review. Some suggested similar work is listed below:
a. Synthesis of Novel Ethoxylated Quaternary Ammonium Gemini Surfactants for Enhanced Oil Recovery Application
b. Imidazolium-based ionic liquids as clay swelling inhibitors: mechanism, performance evaluation, and effect of different anions
c. Machine learning approach to predict the dynamic linear swelling of shales treated with different waterbased drilling fluids
d. Polyoxyethylene quaternary ammonium gemini surfactants as a completion fluid additive to mitigate formation damage
e. Novel gemini surfactant as a clay stabilizing additive in fracturing fluids for unconventional tight sandstones: Mechanism and performance
f. Clay Swelling Mitigation During Fracturing Operations Using Novel Magnetic Surfactants
g. Dicationic surfactants as an additive in fracturing fluids to mitigate clay swelling: A petrophysical and rock mechanical assessment
7. Captions are missing in some of the figures
8. Nomenclature section must be added as per journal policy
9. The spelling of Angle is wrong in many Figures.
10. Why does heading 4 say ‘blood pressure’? Explain
11. The conclusion is also misleading. It is not reflecting what has been done in the paper, Conclusion should restate the research topic. Typically, one sentence can be enough to restate the topic. The conclusion should be clear and concise and state only the most important information
Reviewer 2 Report
The article "Development of Hydrophobic Modified Nanosilica for Pres-sure Reduction and Injection Increase in An Ultra-Low-Permeability reservoir" is an important theme nowadays. The concept of hydrophobization of the surface to protect against material erosion by water is common in many modern materials. In their manuscript, the authors modify SiO2 nanoparticles with an aliphatic amine using the joint coupling agent KH560. According to my quick literature search, the substance, entitled "modified KH560," is still not described in the literature yet. It would be desirable to fully characterize this prepared substance, at least in standard methods such as elemental analysis or HRMS and NMR. Infrared spectra qualitatively documented the modification of SiO2 nanoparticles. Why was it not possible to present the spectrum of the modified KH560 itself? The quantity was then determined from thermogravimetric analysis. The conducted experiments prove the hydrophobization of the surface and its desirable effect on pressure reduction and injection increase. Inhibition of hydration expansion of bentonite due to impregnation with modified nano SiO2 was also demonstrated. Did the authors attempt to quantify the amount of adsorbed modified nanoparticles? For example, by differentiation of concentration of nanoparticles before and after the modification in the modification dispersion?
The presented work may be of interest to readers of applied sciences. I recommend it for publication, but at least the characterization of the modified KH560 must be added. In conclusion, a list of materials could be discussed where the authors assume the use of nanoparticles modified in this way. Furthermore, calculating the costs of preparing modified nanoparticle dispersions would be worth it. It would also be interesting to state how large particles no longer exhibit the described properties due to aggregation or insufficient adsorption capacity.
Reviewer 3 Report
The paper entitled
Development of Hydrophobic Modified Nanosilica for Pres-sure Reduction and Injection Increase in An Ultra-Low-Permeability reservoir
Hao Lai a,b,c, Wei Shi a,b,*, Junqi Wang c, Lei Tang d and Nanjun Lai a,b,c,*
proposes a new kind of anti-wetting and anti-swelling material, based on nanosilica modified with a hydrophobic organic counterpart. The new material is characterized and the results of its application on core rock are presented.
The paper is well written and documented. The experiments prove the effectiveness of the product.
Except for the re-writing of the Abstract, that contains unclarities and repetitions that are annoying, Figure 5 contains a mistake: the y axis is denoted Contact Angel instead of Contact Angle.
I recommend publication, after the revision of the Abstract.
Round 2
Reviewer 1 Report
Thanks for revising the manuscript. I have noticed that spelling mistakes are still not rectified by the authors.
1. Fig 7: the spelling of contact Angle is wrong again.
2. Fig 4: Explain clearly the y-axis of FTIR results, explain the bonds in figure, what peaks and crests are representing.
Reviewer 2 Report
Although the authors reworded the abstract, introduction, and conclusion and improved the formality of their manuscript, they did not respond to my fundamental reservation. The substance "Modified KH560" is still not fully characterized. It is not characterized at all! It is not described in the literature, and if it is, it is not cited. If the authors infer the structure of modified nanoparticles from the structure of this substance, they may make a fundamental mistake. Therefore, I can only accept the manuscript for publication with NMR, HRMS, and/or elemental microanalysis. Alternatively, the authors can reformulate the experimental part as a general description of nanoparticle modification without speculating on their structure. The abstract mentions dodecane, but dodecyl amine was used. Please insert the appropriate characterizations of the substance "modified KH560".The yield could also be listed.
